# 2D-DOA Estimation in Switching UCA Using Deep Learning-Based Covariance Matrix Completion

**DOI:** 10.3390/s22103754

**Published:** 2022-05-14

**Authors:** Ruru Mei, Ye Tian, Yonghui Huang, Zhugang Wang

**Affiliations:** 1National Space Science Center, Chinese Academy of Sciences, Beijing 100190, China; meiruru20@mails.ucas.ac.cn (R.M.); tianye171@mails.ucas.ac.cn (Y.T.); yonghui@nssc.ac.cn (Y.H.); 2University of Chinese Academy of Sciences, Beijing 100049, China

**Keywords:** 2D-DOA estimation, uniform circular array, covariance matrix completion, neural network, deep learning

## Abstract

In this paper, we study the two-dimensional direction of arrival (2D-DOA) estimation problem in a switching uniform circular array (SUCA), which means performing 2D-DOA estimation with a reduction in the number of radio frequency (RF) chains. We propose a covariance matrix completion algorithm for 2D-DOA estimation in a SUCA. The proposed algorithm estimates the complete covariance matrix of a fully sampled UCA (FUCA) from the sample covariance matrix of the SUCA through a neural network. Afterwards, the MUSIC algorithm is performed for 2D-DOA estimation with the completed covariance matrix. We conduct Monte Carlo simulations to evaluate the performance of the proposed algorithm in various scenarios; the performance of 2D-DOA estimation in the SUCA gradually approaches that in the FUCA as the SNR or the number of snapshots increases, which means that the advantages of a FUCA can be preserved with fewer RF chains. In addition, the proposed algorithm is able to implement underdetermined 2D-DOA estimation.

## 1. Introduction

Two-dimensional direction of arrival (2D-DOA) estimation is a critical issue in the domain of array signal processing and has a wide range of applications such as wireless communication [1], sonar [2], radar [3], etc. Meanwhile, the uniform circular array (UCA) is a widely utilized configuration in the context of 2D-DOA estimation due to its attractive abilities, including covering 360° azimuthal space and providing elevation angle information [4,5].

Several algorithms have been proposed to estimate DOAs using UCA such as the UCA-RB-MUSIC algorithm, the UCA-ESPRIT algorithm [6] and the UCA-RARE algorithm [7]. In the above algorithms, it is beneficial to use a large aperture array equipped with multiple synchronized RF chains to avoid spatial aliasing. To strike a good balance between hardware cost and estimation performance, sparse arrays have received significant attention [8,9,10,11,12,13,14]. The coprime array proposed in [8] and the nested array proposed in [10] can obtain O(MN) degrees of freedom (DOFs) with O(M+N) physical sensors. The generalized sum and difference coarray (GSDC) proposed in [12] can obtain O($M^{2}N^{2}$) DOFs with O(M+N) sensors. Sparse nested arrays with a coprime displacement (SNACD) algorithm [14] combine the properties of a nested array and a coprime array, which can simultaneously obtain high-resolution DOA estimation and avoid the mutual coupling influence. However, the above 1D-structured sparse linear arrays (SLAs) cannot perform 2D-DOA estimation.

Furthermore, the switching array equipped with fewer RF chains is a commonly used array structure in the interferometer direction finding systems [15,16]. However, these systems cannot estimate the DOAs of multiple sources simultaneously. To solve this problem, a covariance-matrix-completion-based spatial spectrum estimation method in the switching array was proposed in [17,18]. The shift-invariant matrix completion (SIMC) algorithm proposed in [19] performed DOA estimation by retrieving the missing entries of the sample matrix with reasonable DOA estimation performance. Meanwhile, the SIMC algorithm took advantage of the Toeplitz property of the covariance matrix sampled by a linear array. However, the covariance matrix of UCA did not possess the Toeplitz property. In addition, Ref. [20] proposed a covariance matrix completion method for a nested sparse circular array (NSCA), but it has the limitation that the elevation angle is fixed at 90°.

Strikingly, deep learning (DL) is a data-driven technology that does not rely on pre-assumptions about array geometries and has low online estimation complexity. Hence, DL technology has become a significant tool for DOA estimation [21,22]. In [23,24,25], the DOA estimation problem was treated as a classification problem by discretizing the angular domains in several sectors; however, the classification problem becomes intractable in the context of multiple sources. Additionally, ref. [26] treated DOA estimation as an end-to-end regression neural network (NN) by training five convolutional neural networks (CNNs) with different configurations; yet the number of sources was set to one. Moreover, an NN-based DOA estimator for systems with subarray sampling was presented in [27]. The authors used an NN to estimate the complete covariance matrix from its partial entries; afterwards, the DOA estimation was implemented based on the completed covariance matrix. However, ref. [27] only considered 1D-DOA estimation in scenarios where the number of sources is not greater than the number of RF chains.

As mentioned above, the challenge of performing 2D-DOA estimation in a SUCA has not been well investigated so far. In this paper, we propose a deep feedforward network (DFN)-based matrix completion algorithm to address the 2D-DOA estimation problem in a SUCA. The proposed algorithm is named SUMC (switching UCA matrix completion). The DFN accepts the sample covariance matrix of the SUCA as input, and the output of the DFN is the completed covariance matrix. Afterwards, the MUSIC algorithm is performed for 2D-DOA estimation with the completed covariance matrix. The major contributions of this paper are threefold.

A SUMC algorithm is proposed to estimate the complete covariance matrix from the sample covariance matrix of the SUCA, which means that fewer RF chains than antennas are needed. Therefore, the proposed algorithm can be applied to underdetermined 2D-DOA estimation.A 2D spatial spectrum search scheme based on the classical MUSIC algorithm is performed to estimate the coupled elevation and azimuth angles.Multiple 2D-DOA estimation simulations are carried out to demonstrate that the proposed SUMC-based 2D-DOA estimator (SUMC estimator) is able to preserve the advantage of a FUCA with fewer RF chains.

The rest of the paper consists of four parts. Section 2 formulates the signal model of 2D-DOA estimation in a SUCA. Section 3 presents the proposed DFN framework. Section 4 carries out simulations to demonstrate the predominance of the proposed algorithm. Section 5 concludes the whole paper.

Notations: Vectors are bold lowercase; matrices are bold uppercase. For example, a is a vector and A is a matrix. $A_{i:}$ denotes the i-th row of A; $A_{ij}$ denotes the (i,j)-th entry of A; a[i] denotes the i-th entry of a. Superscripts ${\left(\cdot \right)}^{-1}$, ${\left(\cdot \right)}^{*}$, ${\left(\cdot \right)}^{T}$ and ${\left(\cdot \right)}^{H}$ denote the inverse operation, complex conjugate, transpose and conjugate transpose, respectively. Diag{·} and tr{·} are diagonal matrix and trace operator, respectively. The notation $\langle a,b\rangle =a^{T}b$ represents the inner product of a and b; $∥a∥=\sqrt{\langle a,a\rangle }$ is the $l_{2}$-norm of a; $\left|x\right|$ means taking the absolute value of the real number *x*.

## 2. System Model

Consider *D* uncorrelated far-field narrowband sources from directions ${\left\{\theta_{d},\phi_{d}\right\}}_{d=1}^{D}$ impinging on a the FUCA consisting of $N_{T}$ identical antenna elements distributed over a circle with radius *R*.

The manifold matrix A of the FUCA is given by
(1)$$A=\left[a\left(\theta_{1},\phi_{1}\right),\cdots ,a\left(\theta_{d},\phi_{d}\right),\cdots ,a\left(\theta_{D},\phi_{D}\right)\right]\in C^{N_{T}\times D}$$
where
(2)$$a\left(\theta_{d},\phi_{d}\right)={\left[a_{1}\left(\theta_{d},\phi_{d}\right),\cdots ,a_{n}\left(\theta_{d},\phi_{d}\right),\cdots ,a_{N_{T}}\left(\theta_{d},\phi_{d}\right)\right]}^{T}$$
is the d-th steering vector, and
(3)$$a_{n}\left(\theta_{d},\phi_{d}\right)=e^{j\frac{2\pi R}{\lambda }\sin\theta_{d}\cos\left(\phi_{d}-\gamma_{n}\right)}$$
is the phase difference between the array center and the *n*-th antenna element whose angular coordinate is $\gamma_{n}=\frac{2(n-1)}{N_{T}}\pi$.

Moreover, this system is equipped with $N_{RF}$ ($N_{RF}<N_{T}$) RF chains and a switching network to sample the arriving signals. Notably, only the signals received by $N_{RF}$ selected antennas are sampled at every time instant. As shown in Figure 1, we assume that each RF chain is connected to exactly one antenna element at any time instant; meanwhile, we design the following sampling scheme:$N_{L}$ antennas are locked, which means these antennas are always connected to $N_{L}$ RF chains;$N_{S}$ ($N_{S}=N_{RF}-N_{L}$) antennas are selected and connected to $N_{S}$ RF chains in each switching;The remaining $N_{D}$ ($N_{D}=N_{T}-N_{RF}$) antennas are dumped, which means these antennas are not connected to the RF chain in each switching.

After L ($L=\left(N_{T}-N_{L}\right)/N_{S}$) switchings, all antennas of the SUCA are selected.

The label set of the switching antennas in the l-th switching is $S^{l}≜\left\{p_{1}^{l},p_{2}^{l},\cdots ,p_{N_{S}}^{l}\right\}$; the label set of locked antennas in a SUCA is $L≜\{q_{1},q_{2},\cdots ,q_{N_{L}}\}$; the dump antennas in the l-th switching are labeled as $\left\{v_{1}^{l},v_{2}^{l},\cdots ,v_{N_{D}}^{l}\right\}$. Thus, the label set of switching antennas in a SUCA can be expressed as
(4)$$S=S^{1}\cup \cdots \cup S^{l}\cup \cdots S^{L}.$$

We try to recover the complete covariance matrix of the FUCA by jointly processing K snapshots collected in each switching.

The k-th snapshot in the l-th switching can be written as
(5)$$x^{\left(l\right)}\left(k\right)=G^{\left(l\right)}\left(As^{\left(l\right)}\left(k\right)+\eta^{\left(l\right)}\left(k\right)\right),\;k=1,2,\cdots ,K$$
where $s^{\left(l\right)}\left(k\right)={\left[s_{1}\left(k\right),\cdots ,s_{d}\left(k\right),\cdots ,s_{D}\left(k\right)\right]}^{T}$ denotes the arriving signals, and $\eta^{\left(l\right)}\left(k\right)∼\mathrm{CN}\left(0,\sigma_{\eta }^{2}I_{N_{T}}\right)$ is the additive white Gaussian noise with zero mean and $\sigma_{\eta }^{2}$ variance. The diagonal matrix $G^{\left(l\right)}\in {\left\{0,1\right\}}^{N_{T}\times N_{T}}$ represents the connections between the RF chains and antenna elements in the l-th switching, defined as
(6)$$G_{i,i}^{\left(l\right)}=\left\{\begin{array}{cc} 1, & \mathrm{if}\;i\in L\cup S^{l}\\ 0, & \mathrm{otherwise} \end{array}\right.$$

Hence, the sample matrix received by the l-th switching is written as
(7)$$X_{l}=\left[x^{\left(l\right)}\left(1\right),\cdots ,x^{\left(l\right)}\left(k\right),\cdots ,x^{\left(l\right)}\left(K\right)\right]\in C^{N_{T}\times K}.$$

Afterwards, the *L* sample matrices are packed in X which is given as
(8)$$X=\left[X_{1}\;\vdots \;\cdots \;\vdots \;X_{l}\;\vdots \;\cdots \;\vdots \;X_{L}\right]\in C^{N_{T}\times LK}.$$

Notably, the simultaneously sampled snapshots are able to construct the sample covariance matrix. More specifically, each switching antenna collects only K snapshots, but each locked antenna collects LK snapshots because the locked antenna is selected all the time. Consequently, the sample covariance matrix $\hat{R}$ of the received data can be estimated as
(9a)$$\begin{array}{cc} {\hat{R}}_{i,j} & =\frac{1}{LK}X_{i:}X_{j:}^{H},\;i,j\in L \end{array}$$
(9b)$$\begin{array}{cc} {\hat{R}}_{i,j} & =\frac{1}{K}X_{i:}X_{j:}^{H},\;i,j\in S \end{array}$$
(9c)$$\begin{array}{cc} {\hat{R}}_{i,j} & =\frac{1}{K}X_{i:}X_{j:}^{H},\;i\in L,j\in S \end{array}$$
(9d)$$\begin{array}{cc} {\hat{R}}_{i,j} & =\frac{1}{K}X_{i:}X_{j:}^{H},\;i\in S,j\in L \end{array}$$

For the purpose of illustrating the switching scheme clearly, the SUCA with $N_{T}=10$ antenna elements and $N_{RF}=4$ RF chains connected to $N_{L}=2$ locked antennas and $N_{S}=2$ switching antennas is shown in Figure 2. We lock the center antenna together with the first antenna on the circle, and L=4 switchings are conducted. Thus, the structure of the covariance matrix $\hat{R}$ in this switching scheme is plotted in Figure 3.

Similarly, the k-th snapshot received by the FUCA is written as
(10)$$x_{C}\left(k\right)=As\left(k\right)+\eta \left(k\right),\;k=1,2,\cdots ,K,\cdots ,LK$$
where $\eta \left(k\right)∼\mathrm{CN}\left(0,\sigma_{\eta }^{2}I_{N_{T}}\right)$. Moreover, the complete covariance matrix of the FUCA is calculated as
(11)$${\hat{R}}_{C}=\frac{1}{LK}\sum_{k=1}^{LK}x_{C}\left(k\right)x_{C}{\left(k\right)}^{H}.$$

Additionally, the 2D-DOA estimation can be conducted with the classical MUSIC algorithm as follows
(12)$${\hat{R}}_{C}={\hat{U}}_{s}{\hat{\Lambda }}_{s}{\hat{U}}_{s}^{H}+{\hat{U}}_{n}{\hat{\Lambda }}_{n}{\hat{U}}_{n}^{H}$$
where ${\hat{U}}_{s}$ and ${\hat{U}}_{n}$ are the estimated signal subspace and noise subspace, respectively. Accordingly, the 2D spatial spectrum is formed as
(13)$$f\left(\theta ,\phi \right)=\frac{1}{{∥a\left(\theta ,\phi \right)}^{H}{\hat{U}}_{n}{∥}_{2}^{2}}.$$

We try to recover ${\hat{R}}_{C}$ by retrieving the missing entries of $\hat{R}$. This is a tricky problem because the covariance matrix of UCA does not have the Toeplitz property.

## 3. Deep Feedforward Network for Covariance Matrix Completion

In this section, we design the label and input of the proposed DFN and discuss the network architecture and training strategy.

### 3.1. Data and DFN Architecture

To construct the input data x, we use the real and imaginary parts of the upper triangular elements of the sample covariance matrix $\hat{R}$ and arrange them into a real vector in terms of
(14)x=[R^1,1,R^2,2,⋯,R^NT,NT,ReR^1,2,ImR^1,2,ReR^2,3,ImR^2,3,⋯,ImR^NT−1,NT,⋯,ReR^1,n,⋯,ReR^NT−n+1,NT⋯,ImR^1,NT]∈RNT2
where ReR^i,j and ImR^i,j denote the real part and the imaginary part of ${\hat{R}}_{i,j}$, respectively. The vectorization of the upper triangular part of $\hat{R}$ is shown in Figure 4.

Similarly, the label y is a real vector obtained by the vectorization of the upper triangular part of ${\hat{R}}_{C}$ according to (14).

Our goal is to design a DFN D(·) that can estimate the complete covariance matrix ${\hat{R}}_{C}$ from $\hat{R}$. Given $\hat{R}$, the estimation of ${\hat{R}}_{C}$ can be expressed as
(15)$${\hat{R}}_{\mathrm{DFN}}=D\left(\hat{R};\Theta \right)$$
where Θ denotes the parameters of the network. The values of the parameters Θ are learned via the offline training.

A sketch of the DFN architecture is shown in Figure 5. The DFN consists of one input layer, one output layer and some hidden layers. The main design considerations for network D(·) are to choose the depth of the network and the width of each layer. As a consequence, we choose a fully connected, feedforward network with $N_{h}$ = 5 hidden layers, and each hidden layer is composed of $N_{u}$ = 4096 neurons via experimentations guided by monitoring the estimation error. Moreover, each hidden layer consists of a fully connected (FC) layer and an active function.

In the i-th FC layer, weights and biases are expressed by $w_{i}$ and $b_{i}$, respectively. Furthermore, the output vector of the i-th FC layer ${\hat{y}}_{i}$ is defined as
(16)$${\hat{y}}_{i}=w_{i}x_{i}+b_{i}$$
where $x_{i}$ is the input vector of the i-th FC layer.

Next, the output of the i-th FC layer is sent into the active function. Owing to the nonlinearity of the active function, the DFN can fit arbitrary curves. We employ the Leaky rectified linear unit (LeakyReLU), which is defined as
(17)$$f_{\mathrm{LeakyReLU}}\left(x\right)=\left\{\begin{array}{cc} x, & \mathrm{if}\;x⩾0\\ \lambda x, & \mathrm{otherwise} \end{array}\right.$$
where λ is a small constant. As a result, the output of the i-th hidden layer can be expressed by
(18)$$x_{i+1}=f_{\mathrm{LeakyReLU}}\left({\hat{y}}_{i}\right)$$
where $x_{i+1}$ is the input of the (i+1)-th FC layer.

### 3.2. Training Strategy

The training setup of the DFN is illustrated in Figure 6. In the strategy, x, y and $\hat{y}$ are the input vector, label vector and output vector of the DFN model, respectively.

The cosine similarity (CS) can better measure the similarity of two vectors. Therefore, the loss function is defined as
(19)$$\begin{array}{c} L_{\mathrm{CS}}\left(y,\hat{y}\right)=\left|\frac{\langle y,\hat{y}\rangle }{∥y∥∥\hat{y}∥}-1\right|. \end{array}$$

During training, Θ is updated by an Adam optimizer that runs back propagation based on the loss function in (19). To improve training efficiency, the following techniques are employed. Training data are provided to the model in minibatches, which takes advantage of the high throughput of parallel processing in the graphics processing unit (GPU) and minimizes the latency of memory copying. The learning rate decays exponentially for fine-tuning as the training proceeds.

The training configurations for the DFN are listed in Table 1. A stochastic gradient descent (SGD)-based Adam optimizer is used. The learning rate is initially set to 0.001 and decayed by 0.02 in each epoch.

## 4. Simulation Results

In this section, we conduct numerical experiments to validate the effectiveness of the SUMC estimator.

### 4.1. Simulation Conditions

The considered system consists of $N_{T}=10$ antennas in the form of UCA and $N_{RF}=4$ RF chains. At each time instant, the switching network sequentially selects 4 antennas out of the 10 antennas according to Figure 2, where the UCA antenna elements are numbered counterclockwise, and the center element is marked as 10. In addition, the simulation parameters that are used to generate the training and test datasets are listed in Table 2.

To evaluate the performance of the SUMC estimator for 2D-DOA estimation, the root mean square error (RMSE) is calculated by running Monte Carlo simulations. The RMSE is defined as
(20)$$\mathrm{RMSE}=\sqrt{\frac{1}{WD}\sum_{w=1}^{W}\sum_{d=1}^{D}\left[{\left({\hat{\phi }}_{d}^{w}-\phi_{d}^{w}\right)}^{2}+{\left({\hat{\theta }}_{d}^{w}-\theta_{d}^{w}\right)}^{2}\right]},$$
where W is the total number of Monte Carlo simulations, and $\theta_{d}^{w}$, $\phi_{d}^{w}$, ${\hat{\theta }}_{d}^{w}$ and ${\hat{\phi }}_{d}^{w}$ are the actual elevation angle, actual azimuth angle, estimated elevation angle and estimated azimuth angle of the d-th source in the w-th Monte Carlo simulation, respectively. Moreover, ${\hat{\theta }}_{d}^{w}$ and ${\hat{\phi }}_{d}^{w}$ are obtained by a 2D spatial spectrum search scheme based on the classical MUSIC algorithm. Meanwhile, the direction of *D* sources are randomly generated from the two-dimensional angular space $\left\{\left(\theta_{d},\phi_{d}\right)\;|\;\theta_{d}\in \left(0^{\circ },90^{\circ }\right],\phi_{d}\in \left(0^{\circ },360^{\circ }\right]\right\}$ in each Monte Carlo simulation.

### 4.2. Network Architecture Analysis

In this section, we evaluate the performance of different FC networks for covariance matrix completion. Four datasets were generated in scenarios of D= 1, 2, 3 and 4, respectively. Each individual dataset contains 250,000 sample covariance matrices and corresponding 250,000 complete covariance matrices, of which 80% are used for training, and a subset of the remaining 20% is used for validation. The training data for all FC networks is a mixture of the four datasets. The evolution of the loss on the validation set in the training of different FC networks is shown in Figure 7. In the legend of Figure 7, the *N*-*M* represents the FC network with *N* hidden layers, and each hidden layer is composed of *M* neurons; moreover, 6-2048 × 2-4096 × 4 represents the FC network with six hidden layers. The first two hidden layers consist of 2048 neurons each, and the remaining hidden layers consist of 4096 neurons each.

As shown in Figure 7, the training process starts with a fast fitting followed by a long fine-tuning phase. The loss decreases drastically in the first 40 epochs, then decreases slowly but steadily until hitting a floor. Obviously, the 5-4096 curve is the lowest, and the 2-4096 curve is the highest, indicating that properly expanding and deepening the network can improve its ability to estimate the complete covariance matrix. However, the 6-2048 × 2-4096 × 4 curve is higher than the 5-4096 curve. The main reasons for the performance degradation are gradient instability and network degradation issues in deep networks. Therefore, larger and deeper networks do not necessarily perform better, and the appropriate network needs to be selected according to the specific problem. For the covariance matrix completion problem in this paper, the optimal FC network contains five hidden layers, and each hidden layer consists of 4096 neurons.

Consequently, we trained two DFN models to perform covariance matrix completion in scenarios of D= 1, 2, 3, 4 and the scenario of *D* = 5, respectively.

### 4.3. Comparison to Fully Sampled UCA

In this section, we assess the performance of the SUMC algorithm in scenarios of D= 1, 2 and 3. Additionally, we consider the RMSE of the classical MUSIC algorithm applied to the FUCA, which we use as a lower bound. The RMSE of the classical MUSIC algorithm applied to the FUCA is labeled as FUCA; meanwhile, the RMSE of SUMC algorithm applied to a SUCA is labeled as SUCA.

Figure 8 and Figure 9 depict that the RMSE of the SUMC algorithm decreases considerably as SNR or K increases in scenarios of D= 1, 2 and 3. Apparently, the SUCA curve gradually approaches the FUCA curve as SNR or K increases. Nevertheless, there is a gap between the two curves, and the gap between the SUCA and FUCA curve is becoming wider as D increases. Notably, the proposed algorithm performs worse than the MUSIC algorithm applied to the FUCA because the estimated covariance matrix in (15) is only an approximation of the complete covariance matrix in (11). In summary, the SUMC estimator can perform 2D-DOA estimation and provide reasonable performance.

### 4.4. Comparison to Partial Array

In this section, we compare the performance of the SUMC algorithm and classical MUSIC algorithm applied to a partial array (PA) in the scenario of D= 3. In addition, the layout of the PA is the same as in Figure 2a because a SUCA without matrix completion can use only four elements for DOA estimation. The performance of 2D-DOA estimation is evaluated by RMSE and Success Probability. A successful trial is defined as follows:The number of peaks searched by 2D spatial spectrum search scheme is equal to D;The divation $\left(\Delta \theta_{d},\Delta \phi_{d}\right)$ of each DOA estimate $\left({\hat{\theta }}_{d},{\hat{\phi }}_{d}\right)$ satisfies
$$\Delta \theta_{d}=\left|\theta_{d}-\hat{\theta_{d}}\right|<3^{\circ }\;\&\&\;\Delta \phi_{d}=\left|\phi_{d}-{\hat{\phi }}_{d}\right|<3^{\circ }.$$

Therefore, the Success Probability is expressed as
(21)$$\mathrm{Success}\;\mathrm{Probability}=\frac{W_{success}}{W}$$
where $W_{success}$ is the number of successful trials in each simulation scenario.

As shown in Figure 10a, the estimation accuracy of the three methods is improved as SNR increases, and the RMSE of the proposed method is significantly smaller than that of the MUSIC algorithm applied to PA under low and moderate SNRs. As shown in Figure 10b, it is apparent that the PA curve is far below the SUCA curve. Moreover, the PA curve is always below 40%, and both the SUCA curve and the FUCA curve are almost 100% when SNR ranges from 5 dB to 30 dB. As a result, the effective aperture of the SUCA is enlarged by the SUMC algorithm; meanwhile, the accuracy of DOA estimation is improved.

### 4.5. Resolution of 2D-DOA Estimation

In this section, we evaluate the resolution of the proposed algorithm and the MUSIC algorithm performed in PA when SNR is set to 10 dB and K is set to 1024. The DOAs of two sources in each Monte Carlo trial are selected as follows:The total angular separation of two sources is defined as
$$\varepsilon =\sqrt{\varepsilon_{\theta }^{2}+\varepsilon_{\phi }^{2}};$$The $\varepsilon_{\theta }$ is randomly selected from $\left(0^{\circ },\varepsilon \right)$, $\varepsilon_{\phi }$ is then calculated by
$$\varepsilon_{\phi }=\sqrt{\varepsilon^{2}-\varepsilon_{\theta }^{2}};$$The DOA of the first source is randomly selected from the two-dimensional angular space
$$\left\{\left(\theta_{1},\phi_{1}\right)\;|\;\theta_{1}\in \left(0^{\circ },90^{\circ }-\varepsilon \right],\phi_{1}\in \left(0^{\circ },360^{\circ }-\varepsilon \right]\right\};$$The DOA of the second source is then given by
$$\theta_{2}=\theta_{1}+\varepsilon_{\theta },\phi_{2}=\phi_{1}+\varepsilon_{\phi }.$$

The RMSE for DOAs and the Success Probability for DOA estimation are shown in Figure 11a,b, respectively. As detailed in Figure 11a, for ε as high as $24^{\circ }$, SUCA has an RMSE smaller than $0.48^{\circ }$, whereas PA has an RMSE larger than $1^{\circ }$; moreover, the difference between the PA and the SUCA is most significant at low ε. Figure 11b displays that the Success Probability of the SUCA reaches 90% when $\varepsilon =8^{\circ }$; however, the Success Probability of PA is less than 66% when $\varepsilon <12^{\circ }$. The results in Figure 11 indicate that the SUMC estimator provides better resolution.

### 4.6. Underdetermined 2D-DOA Estimation Problem

In this section, we focus on the capability of the SUMC estimator to solve the underdetermined 2D-DOA estimation problem where $D\geq N_{RF}$. *D* is set to four and five as $N_{RF}=4$. The RMSE in scenarios of four sources and five sources are displayed in Figure 12 and Figure 13, respectively.

As shown in Figure 12 and Figure 13, the performance of the SUMC estimator improves considerably as SNR or K increases in scenarios of four and five sources. In addition, it should be noted from Figure 12 that the RMSE of DOAs in the SUCA approaches that in the FUCA under large K. In detail, Figure 13 depicts that the RMSE of the proposed algorithm is lower than $2.8^{\circ }$ in the scenario of five sources when SNR > 20 dB and K=1024.

Moreover, the spatial spectra obtained from the SUMC estimator are depicted in Figure 14 and Figure 15, where the red circles represent the ground truth value of DOAs. The SNR and K are fixed at 15 dB and 1024, respectively. Figure 14 shows four high and sharp peaks. Significantly, there is almost no deviation between the four peaks and the four red circles. It can be seen from Figure 15 that five peaks have been found by the SUMC estimator; yet there is a slight deviation between the five peaks and the five red circles. On the whole, the proposed algorithm can deal with the underdetermined 2D-DOA estimation problem with acceptable performance under moderate SNR and K.

## 5. Conclusions

In this paper, a 2D-DOA estimation scheme based on covariance matrix completion in a SUCA is proposed. The proposed algorithm starts by using a DFN to complete the sample covariance matrix of a SUCA; then the MUSIC algorithm is performed with the completed covariance matrix. The simulation results indicate that the SUMC estimator enlarges the effective aperture of a SUCA; thereby improving the accuracy, resolution and success probability of 2D-DOA estimation in a SUCA. Furthermore, the results show that the proposed algorithm is able to handle the underdetermined 2D-DOA estimation problem with reasonable performance.

## Figures and Tables

**Figure 1 sensors-22-03754-f001:**
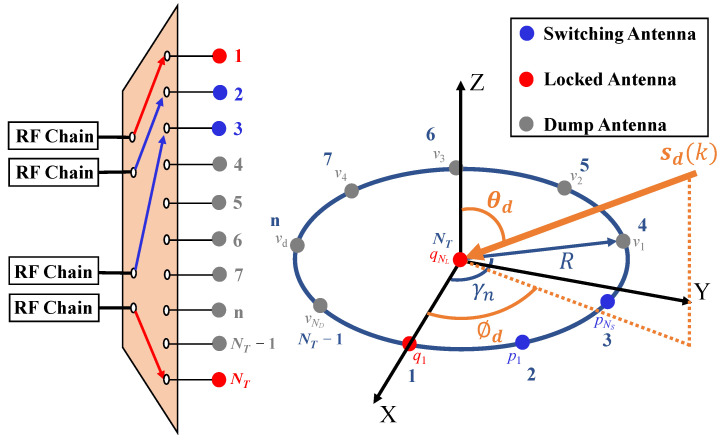
System model.

**Figure 2 sensors-22-03754-f002:**
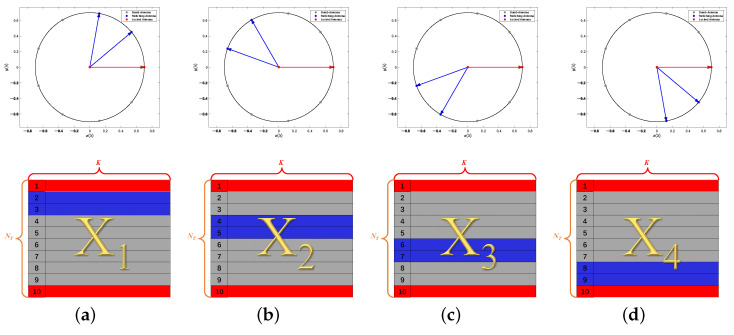
Switching scheme: (**a**) the first switching; (**b**) the second switching; (**c**) the third switching; (**d**) the fourth switching.

**Figure 3 sensors-22-03754-f003:**
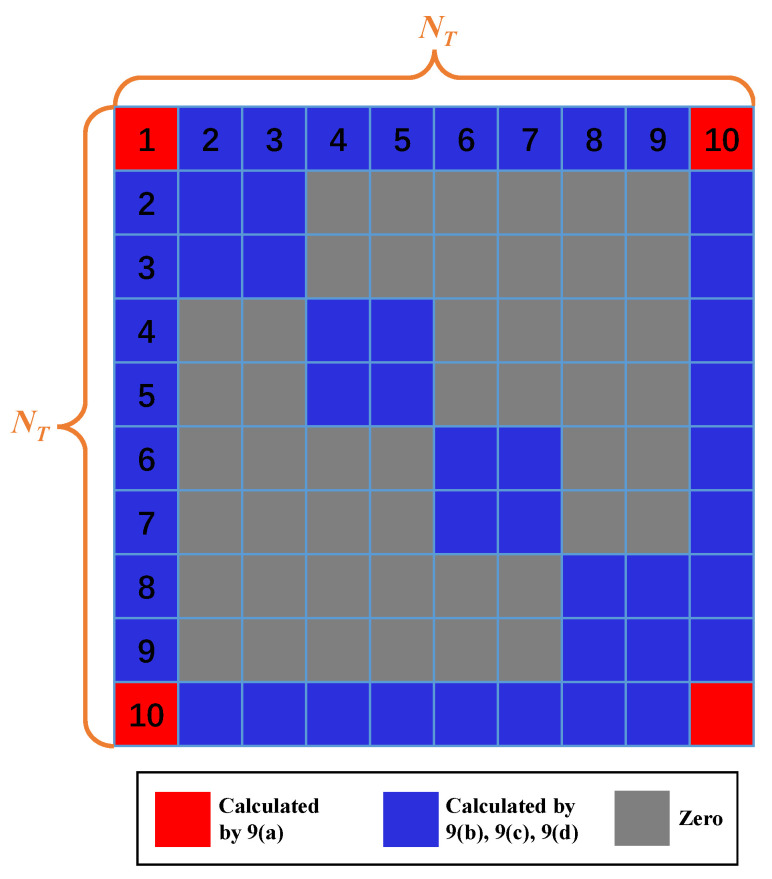
Structure of $\hat{R}$.

**Figure 4 sensors-22-03754-f004:**
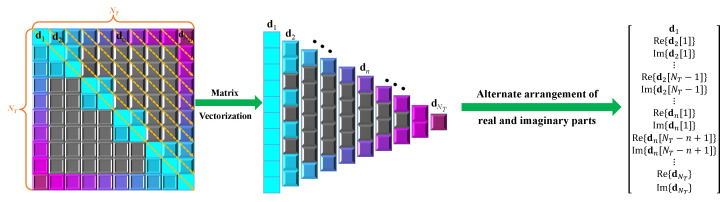
Vectorization of the upper triangular part of $\hat{R}$.

**Figure 5 sensors-22-03754-f005:**
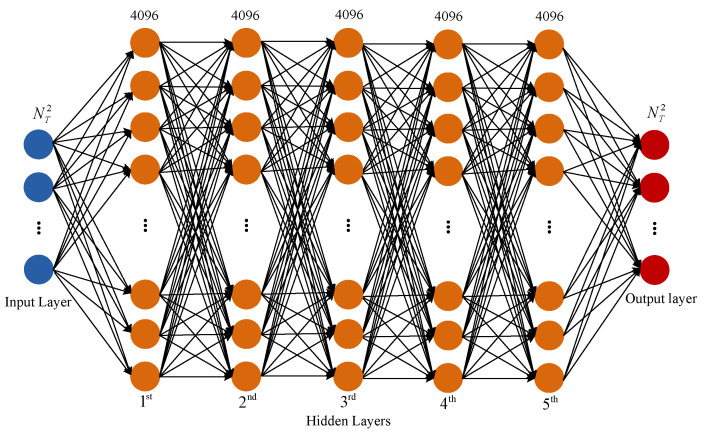
Architecture of DFN.

**Figure 6 sensors-22-03754-f006:**
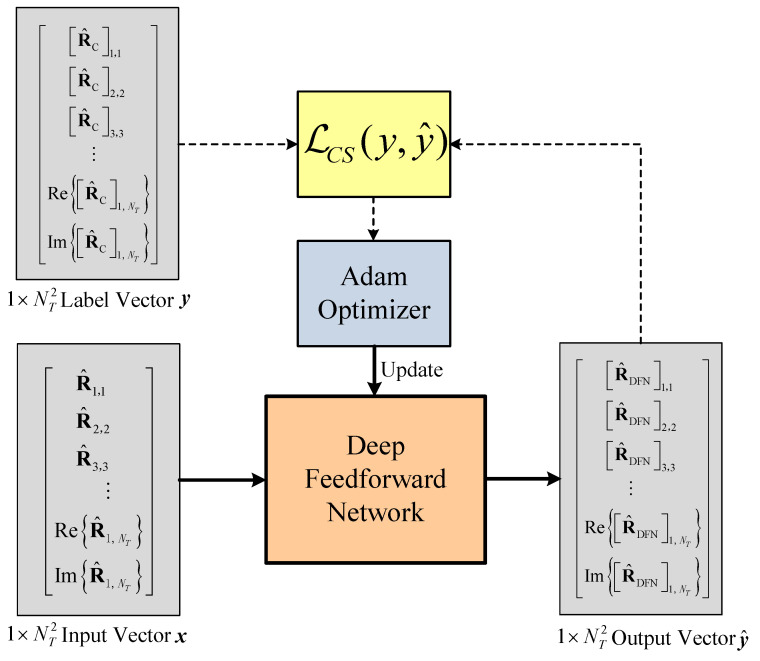
Block diagram of the DFN training system.

**Figure 7 sensors-22-03754-f007:**
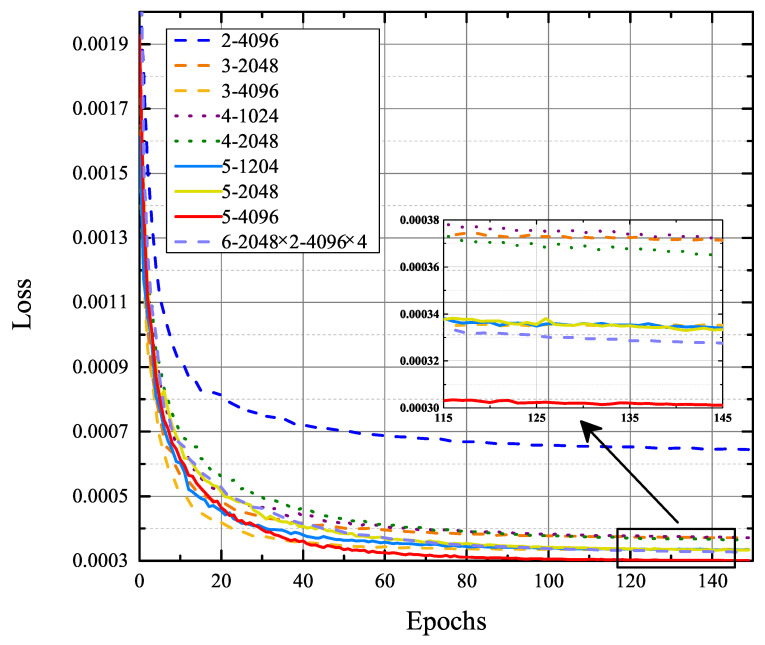
Evolution of loss on the validation set in the training of different FC networks.

**Figure 8 sensors-22-03754-f008:**
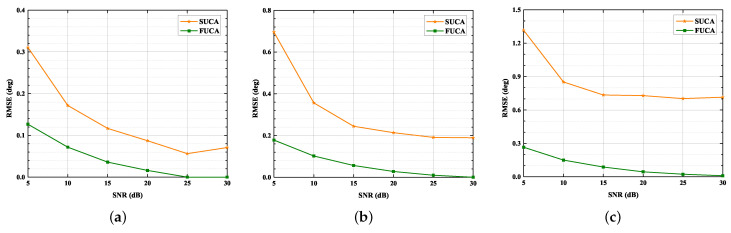
RMSE versus SNR, K = 1024: (**a**) D=1; (**b**) D=2; (**c**) D=3.

**Figure 9 sensors-22-03754-f009:**
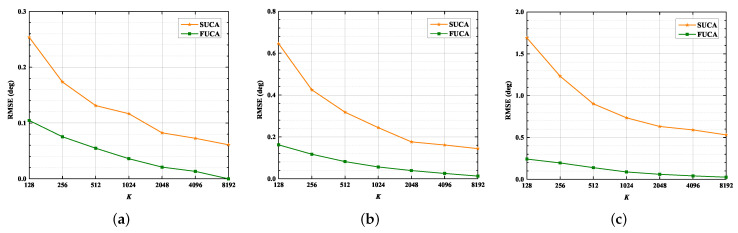
RMSE versus K, SNR = 15 dB: (**a**) D=1; (**b**) D=2; (**c**) D=3.

**Figure 10 sensors-22-03754-f010:**
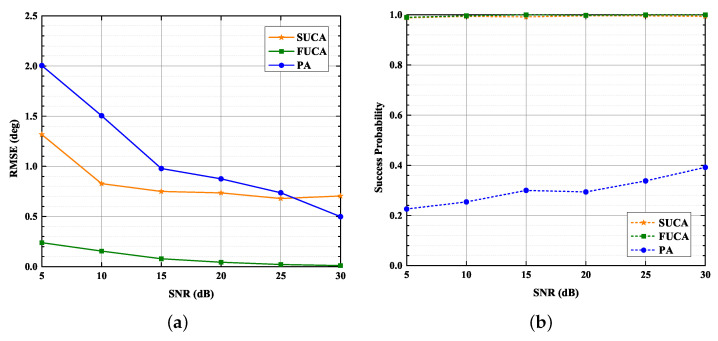
RMSE, Success Probability versus SNR, K=1024, D=3: (**a**) RMSE versus SNR; (**b**) Success Probability versus SNR.

**Figure 11 sensors-22-03754-f011:**
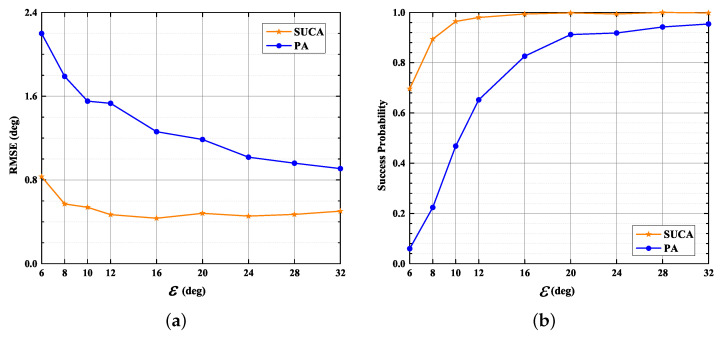
RMSE, Success Probability versus ε, K = 1024, SNR = 10 dB: (**a**) RMSE versus ε; (**b**) Success Probability versus ε.

**Figure 12 sensors-22-03754-f012:**
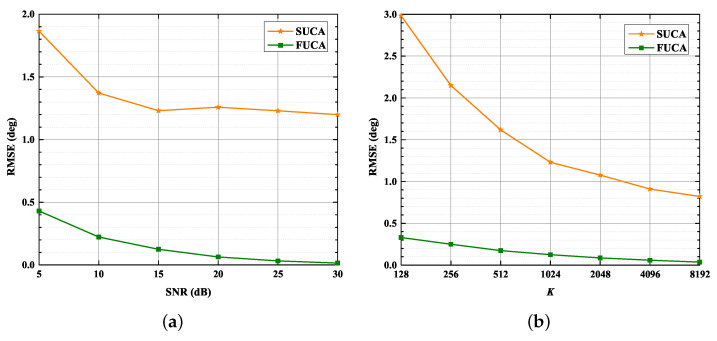
RMSE versus SNR, K: D = 4. (**a**) RMSE versus SNR, K=1024; (**b**) RMSE versus K, SNR = 15 dB.

**Figure 13 sensors-22-03754-f013:**
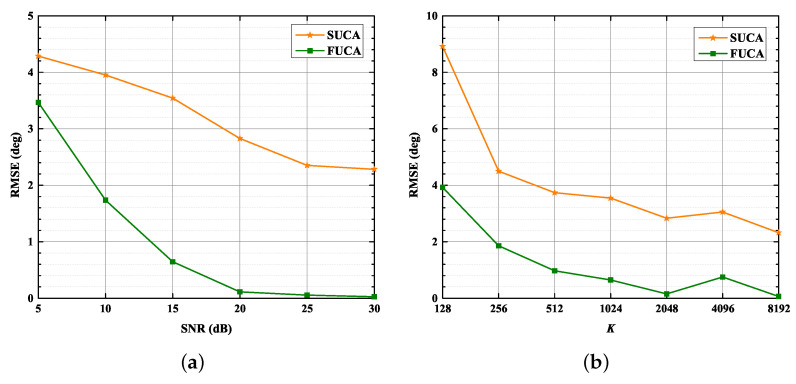
RMSE versus SNR, K. D = 5: (**a**) RMSE versus SNR, K=1024; (**b**) RMSE versus K, SNR = 15 dB.

**Figure 14 sensors-22-03754-f014:**
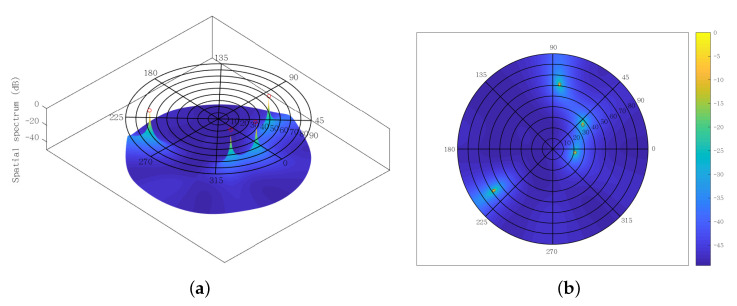
Spatial spectrum using SUMC estimator for solving four sources, SNR = 15 dB, K=1024: (**a**) front; (**b**) top.

**Figure 15 sensors-22-03754-f015:**
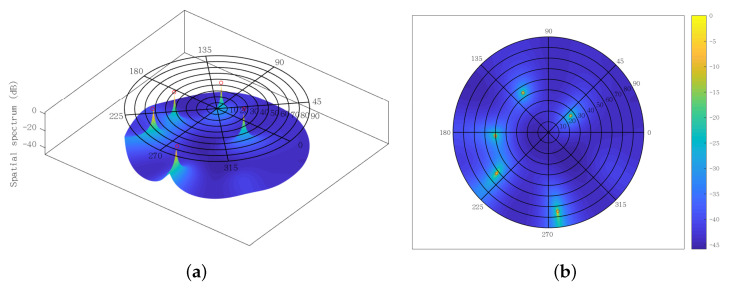
Spatial spectrum using SUMC estimator for solving five sources, SNR = 15 dB, K=1024: (**a**) front; (**b**) top.

**Table 1 sensors-22-03754-t001:** DFN hyperparameters.

Setting	DFN
Maximum Epochs	150
Initial learning rate	$10^{-3}$
Learning rate decay	Exponential, rate 0.02, step one epoch
Optimizer	Adam
Mini-batch size	$256\times 1\times N_{T}^{2}$

**Table 2 sensors-22-03754-t002:** Simulation parameters.

Parameter	Value
Number of antennas $N_{T}$	10
Number of RF chains $N_{RF}$	4
Radius of UCA R	0.7
Number of Monte Carlo trials *W*	500
Number of snapshots K	128, 256, ⋯, 8192
Number of sources D	1, 2, 3, 4, 5
Elevation angle θ	$0^{\circ }:0.1^{\circ }:90^{\circ }$
Azimuth angle ϕ	$0^{\circ }:0.1^{\circ }:360^{\circ }$
SNR	5:5:30 dB

## Data Availability

Not applicable.

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
