# Peer review of "2D-DOA Estimation in Switching UCA Using Deep Learning-Based Covariance Matrix Completion"

_sensors, 2022, doi:10.3390/s22103754_

Round 1

Reviewer 1 Report

The two-dimensional direction of arrival estimation in switching uniform circular array is considered, and a covariance matrix completion method based on neural network is proposed. The paper is well written, but some more explanations about the method are required.

  1. In Fig.5, how to choose the depth of the network and the width of each layer?
  2. In Fig.6, the covariance matrix elements are related to different source number as well as the DOAs, so how to set the training data to cover all the possible source number and DOAs? Besides, how to get the data from FUCA?

Reviewer 2 Report

Major problem:

This FC network is really very large (a lot of weights, comparing to e.g. DL ConvNN)
It means that overfitting is possible.
Please evaluate a lot of different networks (number of layers and number neurons in layer, starting from 2-layers network).

Reviewer 3 Report

This paper adopts DFN to estimate DOA, there are some problems:

  1. Whether the MUSIC algorithm be performed using a NN? Its performance should be better.
  2. Why DFN is the chosen network for DOA estimation? CNN or CRNN has been used in DOA estimation and has good performance. Why not choose them? Some simulations should add to compare with some DNN-based methods.

Round 2

Reviewer 1 Report

The comments have been answered well. 

Author Response

Thanks

Reviewer 2 Report

ok

Author Response

Thanks